# Transcriptome Profiling in the Hippocampi of Mice with Experimental Autoimmune Encephalomyelitis

**DOI:** 10.3390/ijms232314829

**Published:** 2022-11-27

**Authors:** Poornima D. E. Weerasinghe-Mudiyanselage, Sohi Kang, Joong-Sun Kim, Jong-Choon Kim, Sung-Ho Kim, Hongbing Wang, Taekyun Shin, Changjong Moon

**Affiliations:** 1Department of Veterinary Anatomy and Animal Behavior, College of Veterinary Medicine and BK21 FOUR Program, Chonnam National University, Gwangju 61186, Republic of Korea; 2Department of Physiology and Neuroscience Program, Michigan State University, East Lansing, MI 48824, USA; 3Department of Veterinary Anatomy, College of Veterinary Medicine and Veterinary Medical Research Institute, Jeju National University, Jeju 63243, Republic of Korea

**Keywords:** experimental autoimmune encephalomyelitis, gene expression profiling, hippocampus, multiple sclerosis, neuroinflammation, neuroplasticity

## Abstract

Experimental autoimmune encephalomyelitis (EAE), an animal model of multiple sclerosis (MS), approximates the key histopathological, clinical, and immunological features of MS. Hippocampal dysfunction in MS and EAE causes varying degrees of cognitive and emotional impairments and synaptic abnormalities. However, the molecular alterations underlying hippocampal dysfunctions in MS and EAE are still under investigation. The purpose of this study was to identify differentially expressed genes (DEGs) in the hippocampus of mice with EAE in order to ascertain potential genes associated with hippocampal dysfunction. Gene expression in the hippocampus was analyzed by RNA-sequencing and validated by reverse transcription-quantitative polymerase chain reaction (RT-qPCR). Gene expression analysis revealed 1202 DEGs; 1023 were upregulated and 179 were downregulated in the hippocampus of mice with EAE (*p-*value < 0.05 and fold change >1.5). Gene ontology (GO) analysis showed that the upregulated genes in the hippocampi of mice with EAE were associated with immune system processes, defense responses, immune responses, and regulation of immune responses, whereas the downregulated genes were related to learning or memory, behavior, and nervous system processes in the GO biological process. The expressions of hub genes from the search tool for the retrieval of interacting genes/proteins (STRING) analysis were validated by RT-qPCR. Additionally, gene set enrichment analysis showed that the upregulated genes in the hippocampus were associated with inflammatory responses: interferon-γ responses, allograft rejection, interferon-α responses, IL6_JAK_STAT3 signaling, inflammatory responses, complement, IL2_STAT5 signaling, TNF-α signaling via NF-κB, and apoptosis, whereas the downregulated genes were related to synaptic plasticity, dendritic development, and development of dendritic spine. This study characterized the transcriptome pattern in the hippocampi of mice with EAE and signaling pathways underpinning hippocampal dysfunction. However, further investigation is needed to determine the applicability of these findings from this rodent model to patients with MS. Collectively, these results indicate directions for further research to understand the mechanisms behind hippocampal dysfunction in EAE.

## 1. Introduction

Multiple sclerosis (MS) is an autoimmune disease of the central nervous system (CNS) that affects a variety of CNS regions and causes a wide range of symptoms [1]. Hippocampal involvement in MS is evident with severe demyelination [2], neuroinflammation [3], neuronal loss [4], synaptic abnormalities [5], and hippocampal dysfunctions [6]. Previously, accumulating evidence suggested a direct link between neuroinflammation and synaptic abnormalities in the hippocampi of patients with MS [7,8]. For example, increases in the number and activation of microglia cells and complement system coincide with synaptic pruning and the engulfment/elimination of synapses in MS hippocampi [8,9]. Thus, neuroinflammation and synaptic abnormalities in the hippocampus have gained attention as plausible mechanisms underlying the cognitive and emotional symptoms in patients with MS [3,10].

The hippocampus is critical for cognition and emotional regulation [11]. Cognitive and emotional abnormalities in MS are replicated in its most prevalent animal model, experimental autoimmune encephalomyelitis (EAE) [12,13]. These studies have used a battery of behavioral paradigms to assess different aspects of hippocampal function in terms of learning, memory, and emotional behavior. However, the severity of motor impairment in EAE can make the identification of aforementioned non-motor symptoms difficult. Nevertheless, previous studies have used the early pre-symptomatic [14] or late symptomatic phase [15] of EAE in order to explain the validity of functional studies related to behavioral assessment. Thus, as in MS, hippocampal dysfunction occurs in the EAE animal model.

In addition, the EAE animal model mimics the key histopathological (inflammation, demyelination, axonal loss, and gliosis), clinical, and synaptic features of the hippocampus, which are observed during MS pathogenesis [16,17]. Particularly, alterations in synaptic long-term potentiation (LTP) have been found in the hippocampi of mice with EAE [18,19]. Furthermore, in this model, hippocampal dysfunction was observed during the late phase of disease [19,20,21] when neuroinflammation and neurodegeneration are prominent [19,22]. However, the molecular mechanisms underlying hippocampal dysfunction in CNS autoimmune disease, including MS and EAE, remain largely unknown. In the present study, gene expression profiles were analyzed in the hippocampi of mice at the late phase of EAE to gain insights into the underlying mechanisms of hippocampal dysfunction in this animal model.

## 2. Results

### 2.1. Clinical Score following Immunization

Following EAE induction, mice (*n* = 10/group) were monitored daily for the clinical score and body weight until 28 days post-immunization (DPI). Twelve days after immunizing mice with myelin oligodendrocyte glycoprotein peptide 35–55 (MOG_35–55_), a flaccid tail was seen. Hind limb paralysis was observed at approximately 15–24 DPI. By 28 DPI, EAE-affected mice had persistent hind limb paralysis and/or tail atony (F_interaction_ (28, 522) = 27.09, *p* < 0.0001; Appendix A). In line with the EAE onset and progression, the bodyweight of mice with EAE showed a drastic decrease followed by a gradual increment (F_interaction_ (28, 522) = 12.25, *p* < 0.0001; Appendix A).

### 2.2. Differentially Expressed Genes (DEGs) in the Hippocampi of Mice with EAE

To determine the gene expression profile specific to the hippocampus in EAE-induced mice (*n* = 5/group), total RNA-sequencing (RNA-seq) was performed at the late phase (28 DPI). A fold change of 1.5 and a *p*-value of 0.05 were set as cut-off levels for DEG screening. Compared to the corresponding expression levels in the control (CON) group, 1202 genes were recognized as differentially expressed; 1023 were upregulated and 179 were downregulated in the hippocampi of EAE-affected mice. A hierarchical clustering heat map of DEGs is shown in Appendix A.

### 2.3. Functional Analysis of DEGs

During DEG analysis, many genes were significantly different between the CON and EAE-affected hippocampi. Using ShinyGO 0.76 (South Dakota state University, SD, USA), we performed functional annotation analysis to determine the biological significance of the DEGs [23]. Gene ontology (GO) analysis revealed that upregulated genes were associated with biological processes (BP), including the regulation of immune responses, regulation of responses to external stimuli, leukocyte activation, innate immune responses, and regulation of immune response processes (Figure 1A). Figure 1B illustrates the interactions among the pathways in GOBP, where the pathways (nodes) are connected if they share at least 20% of their genes. The darker nodes represent gene sets that are considerably more enriched. Bigger nodes correspond to larger gene sets, whereas thicker edges correspond to more overlapping genes. The search tool for the retrieval of interacting genes/proteins (STRING) analysis illustrates the interaction among the proteins coded by genes represented in the top five pathways in GOBP (Figure 1C). Moreover, cellular components (CC) (Appendix A), including symbiont-containing vacuoles, MHC class I protein complex, inflammasome complex, MHC class I peptide loading complex, and MHC protein complex, and molecular functions (MF) (Appendix A), including peptide antigen binding, MHC protein binding, antigen binding, immune receptor activity, and cytokine receptor activity, were enriched in EAE-affected hippocampi. Appendix A summarizes the detailed report of the GO analysis for upregulated genes in the hippocampi of mice with EAE.

Downregulated genes in EAE-affected hippocampi were associated with BP, including learning, skeletal muscle cell differentiation, olefinic compound metabolic processes, cellular hormone metabolic processes, and embryonic skeletal system development (Figure 2). Moreover, CC, including postsynaptic membrane and synaptic membrane (Appendix A), and MF, including DNA-binding transcription activator activity, DNA-binding transcription activator activity, RNA polymerase II-specific, RNA polymerase II cis-regulatory region sequence-specific DNA binding, cis-regulatory region sequence-specific DNA binding, and transcription cis-regulatory region binding (Appendix A), were downregulated in EAE-affected hippocampi. Appendix A summarizes the detailed report of the GO analysis for downregulated genes in the hippocampi of mice with EAE.

### 2.4. Validation of DEGs in EAE-Affected Hippocampi

To establish whether the gene expressions found by reverse transcription-quantitative polymerase chain reaction (RT-qPCR) were comparable with those discovered by the RNA-seq analyses (*n* = 5/group), we assessed genes chosen based on gene expression and biological significance. Genes were selected from the STRING analysis among the top five GOBP pathways for upregulated genes. There were 28 hub genes whose levels were significantly upregulated in EAE-affected hippocampi among which the qRT-PCR validation was performed for eight genes (Table 1) including chemokine ligand 5 (*Ccl5*), interferon-induced protein with tetratricopeptide repeats 1 (*Ifit1*), tumor necrosis factor alpha (*Tnfα*), C-X-C Motif Chemokine Ligand 10 (*Cxcl10*), TYRO protein tyrosine kinase binding protein (*Tyrobp*), C-C Motif Chemokine Receptor 2 (*Ccr2*), beta-2-microglobulin (*B2m*), and signal transducer and activator of transcription 1 (*Stat1*). However, from the STRING analysis, the node degree of downregulated genes did not reach 10, even in the top ten GOBP pathways. Therefore, four genes with the highest node degree from GOBP, GOCC, and GOMF were selected for RT-qPCR validation (Table 1), including transcription factor jun-B (*Junb*), early growth response 1 (*Egr1*), Fos proto-oncogene (*Fos*), and activity regulated cytoskeleton associated protein (*Arc*). 

A validation of gene expression revealed that in the hippocampi of mice with EAE, the expressions of *Ccl5* (mean (M) = 69.55, standard error of mean (SEM) = 19.65; t(8) = 3.489, *p* = 0.0082), *Ifit1* (M = 3.69, SEM = 0.56; t(8) = 4.551, *p* = 0.0019), *Tnfα* (M = 21.64, SEM = 4.82; t(8) = 4.283, *p* = 0.0027), *Cxcl10* (M = 17.28, SEM = 5.42; t(8) = 2.997, *p* = 0.0172), *Tyrobp* (M = 2.83, SEM = 0.47; t(8) = 3.881, *p* = 0.0047), *Ccr2* (M = 14.26, SEM = 4.13; t(8) = 3.203, *p* = 0.0125), *B2m* (M = 5.73, SEM = 1.00; t(8) = 4.706, *p* = 0.0015), and *Stat1* (M = 3.09, SEM = 0.54; t(8) = 3.851, *p* = 0.0049) genes were significantly increased (Figure 3A). However, the expressions of *Junb* (M = 0.50, SEM = 0.07; t(8) = 4.953, *p* = 0.0011), *Egr1* (M = 0.26, SEM = 0.03; t(8) = 3.842, *p* = 0.0049), *Fos* (M = 0.53, SEM = 0.10; t(8) = 3.520, *p* = 0.0078), and *Arc* (M = 0.60, SEM = 0.16; t(8) = 2.381, *p* = 0.0445) genes were significantly decreased in the hippocampi of mice with EAE (Figure 3B).

### 2.5. Additional Assessment of Functional Differences Using Gene Set Enrichment Analysis (GSEA)

To explore the functional differences of a pre-defined set of genes in the CON vs. EAE-affected groups, we evaluated the differences in the enrichment profiles of hallmark and curated gene sets from the MSigDB database using GSEA, a powerful tool to analyze the expressions of large numbers of genes [24,25]. Notably, the top five positively enriched gene sets in the EAE group were interferon-γ responses, allograft rejection, interferon-α responses, IL6_JAK_STAT3 signaling, and inflammatory responses whereas only four negatively enriched gene sets (Wnt/beta catenin signaling, Hedgehog signaling, estrogen response-early, and myogenesis) were significant (FDR *q*-value < 0.25, NOM *p*-value < 0.05). Moreover, gene sets associated with complement, apoptosis, TNFα signaling via NF-κB, and P13k_AKT_mTOR signaling were also positively enriched in GSEA Hallmarks. Figure 4 and Table 2 provide the detailed enrichment report for hallmark gene sets in the GSEA analysis. 

Additionally, a curated gene set was used to analyze neuroplasticity-related gene set enrichment in the hippocampi of mice with EAE. The results indicated that the most significant gene sets were negatively enriched. The top five negative enrichment of gene sets related to dendritic morphogenesis, dendritic development, CNS neuron development, CNS neuron differentiation, and positive regulation of dendritic development (Figure 5). Table 3 summarizes the detailed enrichment information of the neuroplasticity-related curated gene set.

## 3. Discussion

The present study aimed to analyze gene expression in the hippocampi of mice with EAE in order to unravel plausible molecular changes underlying the hippocampal dysfunction in this CNS autoimmune disease. RNA-seq was used to determine the molecular differences between the hippocampi of CON and EAE-affected mice. Among the DEGs in the hippocampi of mice with EAE, we found that neuroinflammation-related genes were largely upregulated, and a number of genes involved in hippocampal neuroplasticity and learning and memory were downregulated. As a result, this study reports the molecular modifications underlying the hippocampal dysfunctions observed in EAE.

MS is a chronic immune-mediated inflammatory disorder of CNS with diverse clinical symptoms [26,27]. Hippocampal dysfunctions in patients with MS are widely evident with cognitive deficits in all phases of disease progression [28] even though the underpinning mechanisms are yet to be elucidated. In EAE, spatial learning and memory deficits have been reported at different phases of the disease. There is a consensus that synaptic plasticity, the basis of cognitive function, might be influenced by neuroinflammation and immune molecules [29]. In particular, during EAE, abnormalities in synaptic plasticity have been described in the hippocampus [18,19]. However, the underlying molecular mechanisms are still under investigation. Therefore, we attempted to examine the alterations in the gene expression profile of the hippocampus in EAE, which might identify molecules that contribute to hippocampal dysfunction.

In line with most previous reports, we found several inflammation- and immune response-related pathways enriched in the hippocampi of mice with EAE. We confirmed the elevated gene expression levels of inflammatory mediators (cytokines, chemokine ligands, and receptors) in the hippocampi of mice with EAE. Chemokines are involved in immunological processes, including leukocyte recruitment and maturation, and lymphocyte trafficking. For instance, *Cxcl10*, a known chemoattractant for activated T cells [30,31], and *Ccr2* are involved in the microglial phenotype switch to M1 [28]. Chemokines are also known to affect synaptic plasticity and cognitive performance in EAE [32]. For example, increased *Ccl5* paralleled presynaptic defects in EAE [33]. Additionally, the inhibition of *Ccr2* prevented neurobehavioral deficits in a cerebral ischemia model [28]. The major sources of these inflammatory mediators in the hippocampus during the course of EAE seem to be activated microglia and astrocytes, although there is a contribution from blood-borne immune cells [34,35,36,37]. Chronic neuroinflammation, in terms of elevated inflammatory mediators and activated microglia/astrocytes, in the hippocampus at the late phase of EAE is known to affect neuroplasticity, and thus induce cognitive deficits [38,39]. For example, TNFα in the hippocampus triggered astrocyte-mediated synaptic disruption and learning-memory impairment in EAE [37]. Neuroinflammation in the hippocampus is also known to decrease GABAergic neurons, pre-synaptic puncta, and synaptic protein expression [40]. In support of this, brain slices incubated with activated microglia displayed alterations of GABAergic neurotransmission similar to those seen in the EAE brain [38]. The improvement of cognitive function and synaptic plasticity upon the alleviation of neuroinflammation is not surprising because such therapeutic approaches to inhibit microglial/astrogial activation are now being researched. EAE and MS are neuroinflammatory diseases, and it is noteworthy that the targeted inhibition of chemokine pathways in the hippocampus might improve the alterations in hippocampus-related behavior.

In the present study, we observed a significant upregulation of *Stat1* together with interferon inducible gene, *Ifit1*, in the hippocampi of mice with EAE. Moreover, GSEA analysis indicates the upregulation of JAK/STAT signaling in EAE-affected hippocampi. T cell receptor signaling in combination with interferon gamma (IFNγ) stimulation resulted in the activation of *Stat1* [41,42]. *Stat1* and *Ifit1* were highly expressed in peripheral blood mononuclear cells from patients with MS [43] and in CD4^+^ T cells in EAE [44], suggesting that they enhanced IFN signaling. IFNγ-STAT1 signaling determines whether microglia acquire neuroprotective or neurotoxic properties in EAE [45]. In fact, IFNγ and Toll-like receptor signaling pathways were promoted by *Stat1* to shift microglia to an M1 phenotype in EAE [46]. Moreover, *Stat1* is involved in several pathways related to inflammation and synaptic plasticity in neuroinflammatory diseases, including the development, regulation, and termination of immune responses; synaptic plasticity; and the regulation of cognitive function, through the JAK/STAT pathway [47,48]. The inhibition of JAK/STAT1-mediated neuroinflammation alleviated learning and memory impairment, and improved synaptic protein expression in an animal model of vascular dementia [28,49]. Thus, the inhibition of JAK-STAT1 in EAE may improve synaptic and behavioral abnormalities, at least, those related to the hippocampus. However, the applicability of such therapeutic interventions in patients with MS needs to be clarified in the future. 

In addition, we found that *Tyrobp* (*Dap12*) was highly upregulated in the EAE hippocampus. *Tyrobp* plays a major role in transducing activation signals in myeloid cells and in the modulation of genes involved in phagocytosis [50]. The upregulation of *Tyrobp* has also been noted in CNS during acute EAE [51] where it has an important role in the development of autoimmunity during the course of EAE. This is supported by the failure of EAE development in *Tyrobp* knockout (KO) mice [51]. Apart from its role in autoimmunity, *Tyrobp* appears to play role in neuroplasticity because microglial *Tyrobp* KO mice displayed synaptic dysfunction [52]. Clinically, a mutation in *Tyrobp* in humans was associated with presenile dementia and demyelination [53]. However, it remains unclear whether the synaptic effect of microglial *Tyrobp* occurs directly or indirectly. Therefore, increased expression of *Tyrobp* in the present study might be associated with glial cells, and this should be evaluated in the future in order to investigate the synaptic effect of *Tyrobp* in EAE.

The present study showed the upregulation of *B2m* and other genes related to the MHC-I protein complex in the hippocampi of EAE mice. *B2m* is a co-subunit for MHC-I molecules, which are necessary for antigen processing and binding, synaptic processing, and selective elimination [54]. Interestingly, this is the first time the upregulation of MHC-I-related molecules and pathways have been reported in EAE-affected hippocampi. Previously, increased cellular components of the MHC-I complex in the frontal cortex of mice with EAE in the late phase were reported [55]. Furthermore, *B2m* was identified in neuronal and non-neuronal cells in the spinal cord during EAE progression [56,57,58]. In the spinal cord, the upregulation of MHC-I genes correlated with periods of synaptic plasticity disruption by immune cell infiltration [57]. However, the exact effect of MHC-I gene upregulation in the hippocampi of EAE-affected mice is unknown. It is highly possible that *B2m* upregulation underlies synaptic elimination through MHC-I because MHC-I molecules are thought to mediate the selective removal of defective synapses, through activated microglia, in the hippocampi of an epileptic mouse model [59]. Moreover, MHC-I KO mice had increased LTP in the hippocampus [60]. A similar phenomenon also occurred in EAE-affected hippocampi. Thus, this should be a focus for future studies to understand the detailed mechanisms behind the synaptic effects of MHC-I molecules. 

Among the genes downregulated in EAE-affected hippocampi, we identified several immediate early genes (IEGs) including *Junb*, *Egr1*, *Fos*, and *Arc* with higher node degrees in the STRING analysis. For decades, IEGs were used as indirect markers to measure neuronal activity [61]. *Junb*, *Egr1*, and *Fos* belong to an early response family, and *Egr1* is well-known for its role in synaptic plasticity [62]. *Egr1* mediates the expression of a number of late-response genes involved in neuronal processes from growth to plasticity change [61]. The knockdown of *Egr1* in the hippocampus impaired long-term memory [63,64]. *Egr1* regulates the expression of *Arc*, one of the most characterized molecules involved in memory consolidation [65]. Importantly, *Arc* encodes synaptic proteins, and thus is required for the generation of new synapses and plasticity mechanisms [66]. The CNS-specific deletion of *Fos* resulted in the disruption of synaptic plasticity in the hippocampus and learning and memory [67]. Aside from the above, many previous studies reported that the increased expressions of *Egr1, Fos*, and *Arc* were related to an improvement in learning and memory across different animal models of neurological conditions [61,68,69,70], whereas the decreased levels of IEGs in the hippocampus were closely related to mood disorders and cognitive dysfunctions [61]. To the best of our knowledge, this is the first report to show the downregulation of IEGs in the hippocampus in the late phase of EAE. In a previous study, however, increased TNFα and microglial activation were associated with spine loss and decreased levels of *Arc* in the striatum in the early phase of EAE [71]. Taken together, IEGs might be interesting candidates to examine in future studies to understand neuronal activity in the hippocampi of mice with EAE, and the underlying mechanisms of the cognitive and neuropsychiatric symptoms in EAE.

## 4. Materials and Methods

### 4.1. Animals and EAE Induction

Male C57BL/6J mice aged nine weeks (*n* = 10/group) were acquired from Daihan Biolink Co. (Chungbuk, Republic of Korea). The mice were kept in a room with a temperature of 23 ± 2 °C, relative humidity of 50 ± 5%, artificial illumination from 06:00 to 18:00, and 13–18 air volume changes per hour. All mice had free access to water and standard rodent food (Samyang Feed; Republic of Korea). All experimental and animal handling procedures were carried out in accordance with the guidelines of the institutional care and use committee of Chonnam National University (12 August 2022; CNU IACUC-YB-2022-101), and animal care adhered to internationally agreed standards for laboratory animal use and care, as mandated by the National Institutes of Health (NIH). Every attempt was made to reduce the number of animals utilized and their suffering. 

EAE was induced in mice (*n* = 10) as previously described [72]. Briefly, the EAE group was immunized with 1 mg/mL of MOG_35–55_ peptide (purity > 96.44%; #051716, GL Biochem Ltd., Shanghai, China) emulsified in complete Freund’s adjuvant (CFA; #F5881, Sigma-Aldrich, St. Louis, MO, USA) supplemented with 5 mg/mL of Mycobacterium tuberculosis H37Ra (#BD231141, Difco Laboratories Inc., Franklin Lakes, NJ, USA) by subcutaneous injection into the hind flank. At days 0 and 2 post-immunization (DPI), 500 ng of pertussis toxin (List Biological Laboratories, Inc., Campbell, CA, USA) was injected intraperitoneally into mice. Control animals remained non-immunized as previously described [73]. Following immunization, the mice were weighed daily and scored clinically as follows: grade 0 (G.0), no signs; G.1, floppy tail; G.2, mild paraparesis; G.3, severe paraparesis; G.4, tetraparesis; and G.5, moribund or death. At 28 DPI, hippocampal tissues from control and EAE mice were sampled for RNA-Seq and RT-qPCR validation. 

### 4.2. RNA Isolation and RNA-Seq

Mice (*n* = 5/group) were decapitated and their hippocampi were removed. Following the manufacturer’s instructions, total RNA was extracted using an RNeasy^®^ Mini Kit (#74106, Qiagen, Hilden, Germany). Quant-IT RiboGreen (#R11490, Invitrogen, Carlsbad, CA, USA) was utilized to determine the total RNA concentration. Samples were tested on the TapeStation RNA ScreenTape (#5067-5576, Agilent Technologies, Palo Alto, CA, USA) to determine the integrity of the total RNA. For RNA library construction, only high-quality RNA preparations with an RNA integrity number greater than 7.0 were employed.

A library was constructed separately with 1 μg of total RNA for each sample by the Illumina TruSeq Stranded mRNA Sample Prep Kit (RS-122-2101, Illumina, Inc., San Diego, CA, USA). The initial stage in the procedure is to use poly-T-attached magnetic beads to purify the poly-A carrying mRNA molecules. Following purification, the mRNA is fragmented into small pieces under increased temperature using divalent cations. SuperScript II reverse transcriptase (#18064-014, Invitrogen, Waltham, MA, USA) and random primers are utilized to convert the cleaved RNA fragments into first strand cDNA. The second strand of cDNA is then synthesized using DNA polymerase I, RNase H, and dUTP. These cDNA fragments are subsequently subjected to an end repair procedure, the insertion of a single ‘A’ base, and adapter ligation. To construct the final cDNA library, the products are purified and enriched by PCR. The libraries were quantified utilizing KAPA Library Quantification kits for Illumina Sequencing platforms in accordance with the qPCR Quantification Protocol Guide (KK4854, Kapa Biosystems, Wilmington, MA, USA) and qualified utilizing the TapeStation D1000 ScreenTape (#5067-5582, Agilent Technologies, Palo Alto, CA, USA). Indexed libraries were then sent to Illumina NovaSeq (Illumina, Inc.), and Macrogen Inc. (Seoul, Republic of Korea) completed the paired-end sequencing. 

### 4.3. DEGs, Enrichment Analysis, and Protein–Protein Interaction Analysis

The Bowtie 2 tool was used to match the trimmed reads to the indexed genome. Cufflinks [74] was used to determine fragments per kb per million reads (FPKM). Genes with log_2_ fold change more than 0.75 and *p*-value less than 0.05 were declared differentially expressed. ShinyGO 0.76 (South Dakota state University, SD, USA; http://bioinformatics.sdstate.edu/go/; accessed on 13 August 2022) was used to examine the gene ontology enrichment of the DEGs [23]. Three GO datasets, including BP, CC, and MF, were analyzed, respectively for upregulated and downregulated genes [75]. An interactive plot was created to show the interactions between the top ten enriched pathways, where two pathways (nodes) were connected if they shared at least 20% (default) of their genes, darker nodes indicated more significantly enriched gene sets, bigger nodes reflected larger gene sets, and thicker edges represented more overlapping genes. The genes under the top ten GO functional categories at a false discovery rate (FDR) cutoff at 0.05 were analyzed further for possible interactions. Using the STRING database, a protein–protein interaction network was created to determine the potential physical or functional interactions between DEGs and to identify the hub genes. For the minimum needed interaction score, the highest confidence of ≥0.900 and medium confidence of ≥0.400 were set for upregulated and downregulated DEGs, respectively. Each node in the STRING analysis corresponded to the protein/gene product and the edges represent evidence for associations.

Additionally, GSEA version 4.2.3 (www.gsea-msigdb.org) was employed to analyze the gene sets enriched in the RNA-seq data. GSEA is a useful statistical approach for identifying significantly enriched or depleted groups of genes [24]. In the above comparisons, analysis parameters were set up following the GSEA user guide. An enriched dataset expression matrix using the Hallmark gene sets was created at the cut-off levels FDR *q*-value < 0.25 and NOM *p*-value < 0.05 [24]

### 4.4. RNA Extraction, cDNA Synthesis, and RT-qPCR

RNA extraction, complementary DNA (cDNA) synthesis, and RT-qPCR were conducted in accordance with our previous studies [76]. Briefly, cDNA was made using the SuperiorScript III cDNA synthesis kit (#EZ405S, Enzynomics, Daejeon, Republic of Korea). The cDNA was diluted with RNase-free water to a final concentration of 8 ng/µL, and the samples were kept at −80 °C. RT-qPCR was carried out using TOPreal^TM^ SYBR Green qPCR PreMix (#RT500M, Enzynomics, Daejeon, Republic of Korea) and the LineGene 9600 Plus machine (BIOER, Hangzhou, China) following the manufacturer’s instructions. The primers for RT-qPCR are shown in Table 1. The annealing temperature for the reaction was 58 °C, and the built-in software created the amplification curves and calculated the threshold cycle values. The GAPDH reference gene was used to normalize all the readouts. Data were reported as the mean relative values compared to the CON group using the 2^−∆∆CT^ method.

### 4.5. Statistical Analysis

The RT-qPCR findings were analyzed to determine any differences between the CON and EAE-affected groups using independent two-tailed Student *t*-tests. Clinical score and body weight changes between the CON and EAE-affected groups were compared using two-way ANOVA followed by Sidak’s multiple comparisons test. All statistical analyses were performed by GraphPad (version 9.3.1, GraphPad Software, San Diego, CA, USA), and all data are presented as the mean (M) ± standard error of mean (SEM). A *p*-value less than 0.05 was considered statistically significant in all the analyses.

## 5. Conclusions

The present study strengthens the available evidence for hippocampal neuroinflammation, even in the chronic phase of EAE in mice. Importantly, for the first time, we unraveled the significant downregulation of IEGs in the hippocampi of mice with EAE, which may underlie hippocampus-related behavioral and synaptic dysfunctions in this animal model. Furthermore, these findings might explain the alterations in hippocampus-related symptoms in patients with MS and therefore should be confirmed in future studies.

## Figures and Tables

**Figure 1 ijms-23-14829-f001:**
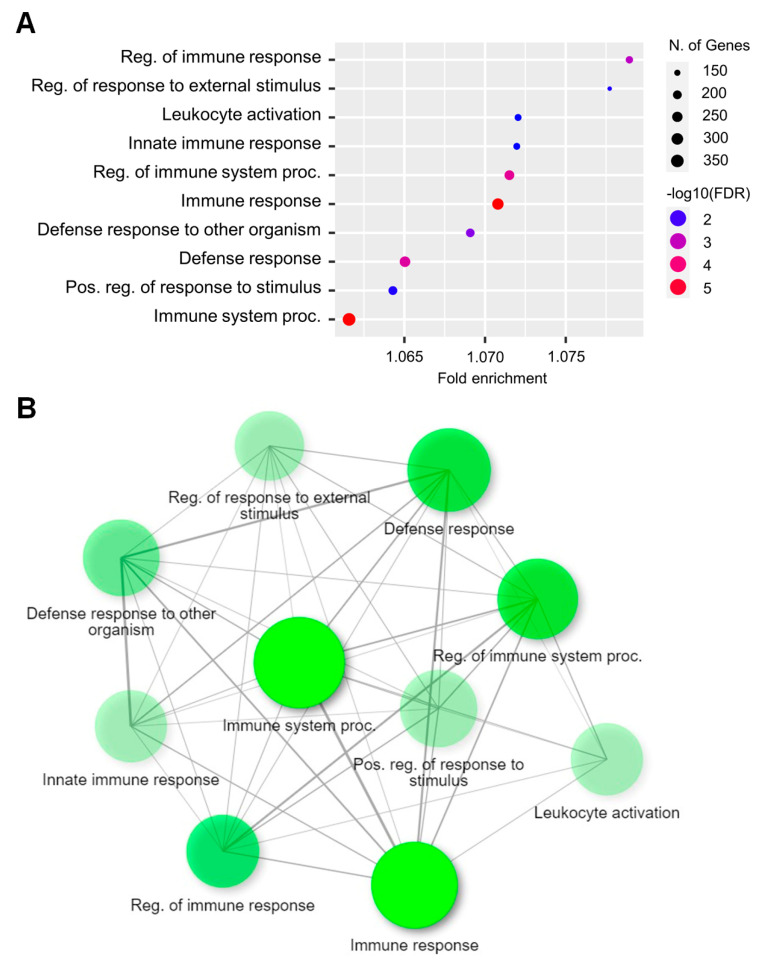
Gene ontology (GO) analysis of upregulated genes from the RNA-seq data. (**A**) Dot plot of enriched genes under GOBP (top ten) in EAE-affected hippocampi with >1.5-fold change and *p-*value < 0.05. (**B**) Interactive plot showing the relationship between enriched pathways. (**C**) STRING analysis of the top five pathway genes in GOBP at the highest confidence level (0.9).

**Figure 2 ijms-23-14829-f002:**
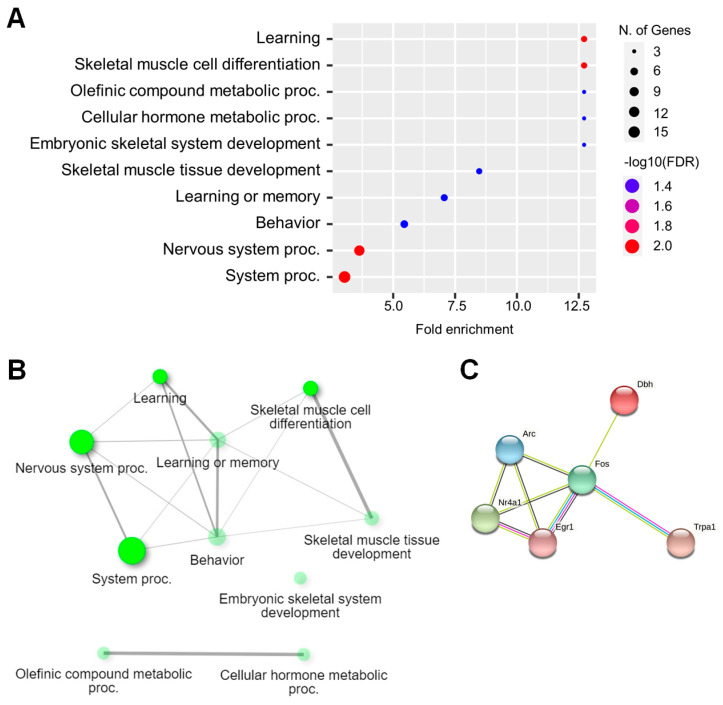
Gene ontology (GO) analysis of downregulated genes from the RNA-seq data. (**A**) Dot plot of enriched genes under GOBP (top ten) in EAE-affected hippocampi with > −1.5-fold change and *p-*value < 0.05. (**B**) Interactive plot showing the relationship between enriched pathways. (**C**) STRING analysis of the top ten pathway genes in GOBP at a medium confidence level (0.4).

**Figure 3 ijms-23-14829-f003:**
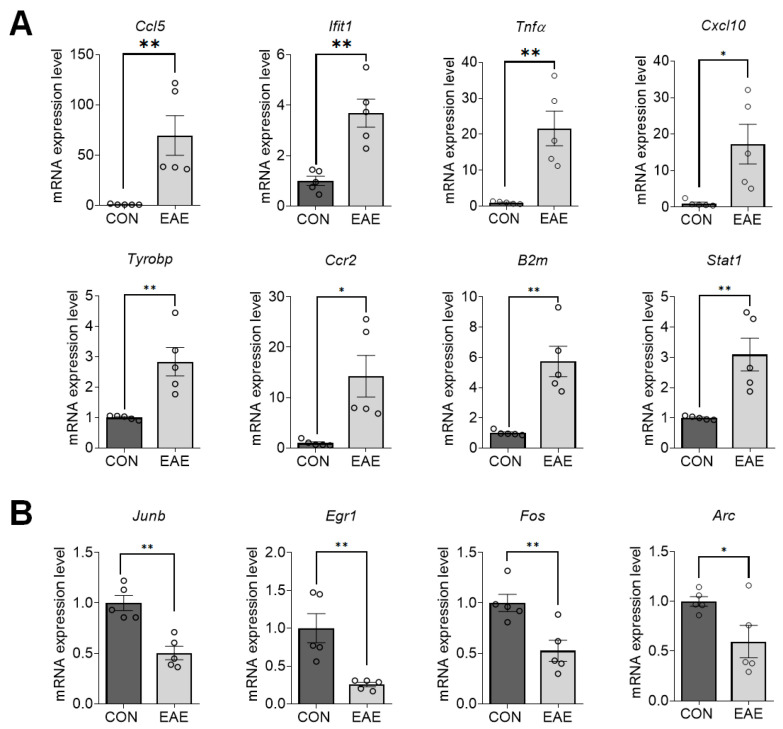
Changes in 12 selected differentially expressed genes (DEGs) in the hippocampi of mice with EAE. The bar graphs show the relative expression levels of upregulated (**A**) and downregulated (**B**) genes from RNA-seq data. Data are expressed as the mean ± SEM (*n* = 5/group). * *p* < 0.05 and ** *p* < 0.01 vs. the CON group.

**Figure 4 ijms-23-14829-f004:**
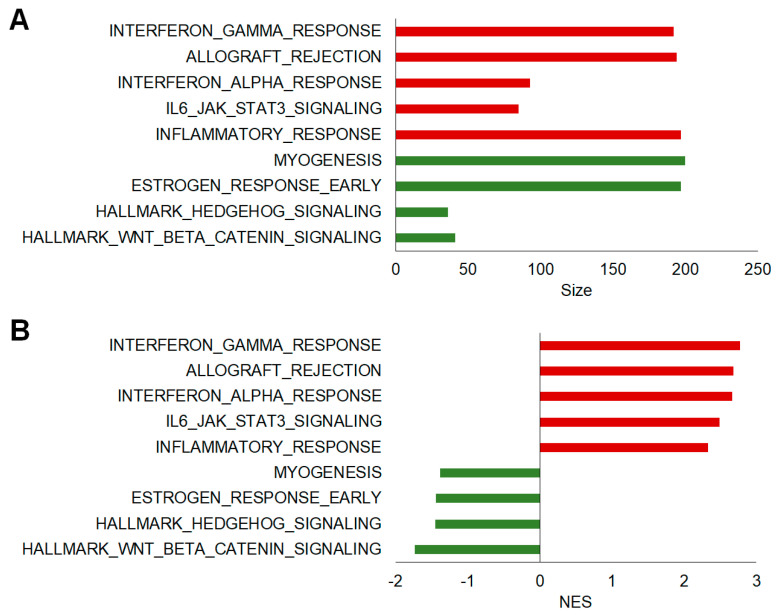
Hallmark gene sets enriched in the hippocampi of mice with EAE. The bar graphs show the top GSEA hallmark gene sets (FDR *q*-value < 0.25, NOM *p*-value < 0.05) that were enriched (positively (red) and negatively (green)) in EAE. The bars are ordered by the size of the gene set (**A**) and the normalized enrichment score (NES) indicates the strength of the enrichment (**B**).

**Figure 5 ijms-23-14829-f005:**
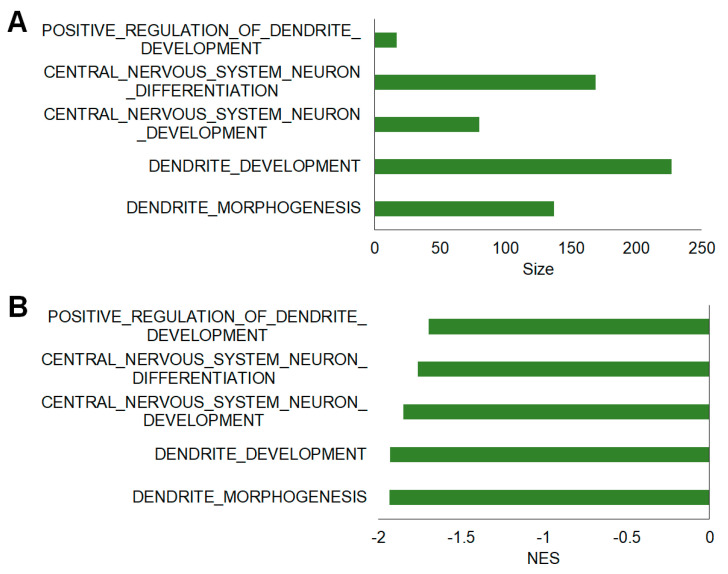
Neuroplasticity-related curated gene sets enriched in the hippocampi of mice with EAE. The bar graphs show the top five gene sets (FDR *q*-value < 0.25, NOM *p*-value < 0.05) that were negatively enriched in the EAE-affected group. The bars are ordered by the size of the gene set (**A**) and the normalized enrichment score (NES) indicates the strength of the enrichment (**B**).

**Table 1 ijms-23-14829-t001:** Summary of RNA sequencing results for selected differentially expressed genes (DEGs) for RT-qPCR validation.

Gene Symbol	FC–RNAseq	FC–RT-qPCR	NCBI Sequence	Primer Pair (5′-3′)	Length (bp)
*Ccl5*	20.954 *	69.55 **	NM_013653.3	F-CAATCTTGCAGTCGTGTTTGTC R-AGGGGATTACTGAGTGGCATC	197
*Ifit1*	3.945 *	3.688 **	NM_008331.3	F-TACAGCAACCATGGGAGAGAATG R-ACTGGACCTGCTCTGAGATT	143
*Tnfα*	16.693 *	21.64 **	NM_013693.3	F-CCCAAAGGGATGAGAAGTTCC R-TGGGCTACAGGCTTGTCACTC	109
*Cxcl10*	13.65 *	17.28 *	NM_021274.2	F-CCACGTGTTGAGATCATTGCC R-GAGGCTCTCTGCTGTCCATC	184
*Tyrobp*	2.8097 *	2.832 **	NM_011662.3	F-TTAAGTCCCGTACAGGCCCA R-TTGTTTCCGGGTCCCTTCCG	170
*Ccr2*	19.534 *	14.26 *	NM_009915.2	F-AGGAGCCATACCTGTAAATGCC R-ATGCCGTGGATGAACTGAGG	163
*B2m*	5.464 **	5.728 **	NM_009735.3	F-CTCACACTGAATTCACCCCC R-TCACATGTCTCGATCCCAGTAG	300
*Stat1*	2.803 *	3.091 **	NM_001357627.1	F-GCCTCTCATTGTCACCGAAGAAC R-TGGCTGACGTTGGAGATCACCA	100
*Junb*	−1.7655 ***	−1.982 **	NM_008416.3	F-GGATCCCTATCGGGGTCTCA R-TTGCTGTTGGGGACGATCAA	156
*Egr1*	−2.397 ***	−3.915 **	NM_007913.5	F-GCACCTGACCACAGAGTCCTTT R-GGCCACTGACTAGGCTGAAAA	188
*Fos*	−2.3463 ***	−1.898 **	BC029814	F-GGGCTGCACTACTTACACGT R-TGCCTTGCCTTCTCTGACTG	169
*Arc*	−1.94 **	−1.676 *	NM_018790.3	F-GATCTTTCCTGCTGTGCCCT R-CGCAACAAGGCCTACTCAGA	109
*Gapdh*			NM_008084	F-CATCACTGCCACCCAGAAGACTG R-ATGCCAGTGAGCTTCCCGTTCAG	153

**Abbreviations:***Arc*, activity regulated cytoskeleton associated protein; *B2m*, beta-2-microglobulin; *Ccl5*, chemokine ligand 5; *Ccr2*, C-C motif chemokine receptor 2; *Cxcl10*, C-X-C motif chemokine ligand 10; *Gapdh*, glyceraldehyde-3-phosphate dehydrogenase; *Egr1*, early growth response 1; F, forward; FC, fold change; *Ifit1*, interferon induced protein with tetratricopeptide repeats 1; *Junb*, transcription factor jun-B; NCBI, National Center for Biotechnology Information; R, reverse; Stat1, signal transducer and activator of transcription 1; *Tnfα*, tumor necrosis factor alpha; *Tyrobp*, TYRO protein tyrosine kinase binding protein. * *p* < 0.05, ** *p* < 0.01, *** *p* < 0.001 (CON vs. EAE).

**Table 2 ijms-23-14829-t002:** Gene Set Enrichment Analysis (GSEA) results according to the MSigDB Hallmark gene sets.

MSigDB Gene Set	Size	NES	FDR *q*-value	NOM *p*-value
**Negatively enriched gene sets**				
Wnt_beta-catenin signaling	41	−1.74	0.015	0.008
Hedgehog signaling	36	−1.45	0.06	0.033
Estrogen response early	197	−1.44	0.05	0
Myogenesis	200	−1.38	0.06	0
**Positively enriched gene sets**				
Interferon gamma response	192	2.78	0	0
Allograft rejection	194	2.68	0	0
Interferon alpha response	93	2.67	0	0
Il6_jak_stat3 signaling	85	2.49	0	0
Inflammatory response	197	2.33	0	0
Complement	193	2.24	0	0
Il2_stat5 signaling	199	2.09	0	0
Kras signaling up	199	1.85	0	0
E2f targets	200	1.82	0	0
G2m checkpoint	196	1.81	0	0
Coagulation	138	1.79	0	0
Apoptosis	161	1.77	0	0
Tnfα signaling via nfκb	198	1.62	0.005	0
P13k_akt_mtor signaling	105	1.35	0.096	0.041

**Abbreviations:** FDR, false discovery rate; NES, normalized enrichment score; NOM, nominal.

**Table 3 ijms-23-14829-t003:** Gene Set Enrichment Analysis (GSEA) results according to the MSigDB neuroplasticity-related curated gene sets.

MSigDB Gene Set	Size	NES	FDR *q*-value	NOM *p*-value
Dendrite morphogenesis	137	−1.933	0.01	0
Dendrite development	227	−1.929	0.01	0
Central nervous system neuron development	80	−1.849	0.01	0
Central nervous system neuron differentiation	169	−1.763	0.02	0
Positive regulation of dendrite development	17	−1.698	0.02	0.01
Dendritic spine morphogenesis	57	−1.694	0.02	0.01
Regulation of dendrite development	97	−1.686	0.02	0
Regulation of dendrite morphogenesis	63	−1.667	0.02	0.01
Dendritic spine development	89	−1.591	0.03	0.01
Regulation of dendritic spine morphogenesis	43	−1.578	0.03	0.02
Dendrite terminus	12	−1.574	0.03	0.03
Regulation of dendrite extension	25	−1.551	0.03	0.02
Dendrite extension	35	−1.524	0.04	0.03
Positive regulation of dendrite morphogenesis	34	−1.502	0.04	0.01

**Abbreviations:** FDR, false discovery rate; NES, normalized enrichment score; NOM, nominal.

## Data Availability

This paper utilized original data not used in other publications. The datasets generated and/or analyzed in the present study are available from the corresponding author upon reasonable request.

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
