# Peer review of "Transcriptome Profiling in the Hippocampi of Mice with Experimental Autoimmune Encephalomyelitis"

_ijms, 2022, doi:10.3390/ijms232314829_

Round 1
Reviewer 1 Report
In this interesting paper, Weerasinghe-Mudiyanselage et al have characterised the hippocampal transcriptome in EAE. Specifically they show a downregulation of intermediate early genes, and correlate this to cognitive and synaptic alterations associated with this model.
Some minor alterations:
1) Please add a sentence in the methods to describe how control animals were immunised.
2) Please add in the time point of sacrifice to the main methods.
3) Clarify why only n=5 were used for RNA isolation/sequencing when n=10 were immunised.
Author Response
Comment 1. Please add a sentence in the methods to describe how control animals were immunized.
Response: We sincerely appreciate the reviewer’s comment. As suggested, we have added a sentence to explain the establishment of the control group.
(Sub-section 4.1, Line 18)
“Control animals remained non-immunized as previously described [73].”
Comment 2. Please add in the time point of sacrifice to the main methods.
Response: As suggested, we have added a sentence to state the time point of sampling in the Materials and Methods section.
(Sub-section 4.1, Lines 21–22)
“At 28 DPI, hippocampal tissues from control and EAE mice were sampled for RNA-Seq and RT-qPCR validation.”
Comment 3. Clarify why only n = 5 were used for RNA isolation/sequencing when n = 10 were immunized.
Response: In this study, only 5 samples per group were utilized for the RNA-seq analyses while the remaining 5 samples per group were used for RT-qPCR to validate the results from RNA-seq.
Reviewer 2 Report
1. How many reads and how many features are identified for this study?
2. In the paragraph on page 2, “2.2. Differentially Expressed Genes (DEGs) in the Hippocampi of Mice with EAE”, why the p value is used for DEG cutoff, instead of FDR? However, in the method section, page 14, section4.3, the DEG is described as “. Genes with log2 fold change more than 0.75 and false discovery rate (FDR) less than 5% (FDR < 0.05) were declared differentially expressed.”, what are the difference between them?

Author Response
Comment 1. How many reads and how many features are identified for this study?
Response: In this study, approximately 61.4–89.4 million reads for each sample were generated and 40202 features in total were identified as raw data. We append the detailed read counts in each sample, as follows:
Sample ID |
Total reads |
Con1 |
63,339,846 |
Con2 |
68,106,906 |
Con3 |
86,893,144 |
Con4 |
87,629,674 |
Con5 |
81,992,512 |
EAE1 |
70,083,862 |
EAE2 |
89,411,148 |
EAE3 |
84,665,672 |
EAE4 |
61,376,162 |
EAE5 |
86,590,182 |
Comment 2. In the paragraph on page 2, “2.2. Differentially Expressed Genes (DEGs) in the Hippocampi of Mice with EAE”, why the p value is used for DEG cutoff, instead of FDR? However, in the method section, page 14, section4.3, the DEG is described as “. Genes with log2 fold change more than 0.75 and false discovery rate (FDR) less than 5% (FDR < 0.05) were declared differentially expressed.”, what are the difference between them?
Response: We sincerely appreciate the reviewer’s comment. DEGs were selected depending on both log2 fold change (log2 FC > 0.75) and the p-value (p < 0.05). We have corrected the misinterpretation of this selection criteria in the Materials and Methods section in the manuscript, as follows:
(Sub-section 4.3, Lines 2–4)
“Genes with log2 fold change more than 0.75 and p-value less than 0.05 were declared differentially expressed.”
Round 2
Reviewer 1 Report
None
